# State-of-the-Art Report: The Self-Healing Capability of Alkali-Activated Slag (AAS) Concrete

**DOI:** 10.3390/ma16124394

**Published:** 2023-06-14

**Authors:** Nancy Hammad, Amr Elnemr, Ibrahim G. Shaaban

**Affiliations:** 1Civil Engineering Program, German University in Cairo (GUC), Cairo 11835, Egypt; nancy.saad@guc.edu.eg; 2Civil Engineering Department, Sherbrooke University, Sherbrooke, QC J1K 2R1, Canada; 3School of Computing and Engineering, University of West London, London W5 5RF, UK

**Keywords:** alkali-activated slag, self-healing, drying shrinkage, autogenous healing, autonomous healing, crack widths

## Abstract

Alkali-activated slag (AAS) has emerged as a potentially sustainable alternative to ordinary Portland cement (OPC) in various applications since OPC production contributed about 12% of global CO_2_ emissions in 2020. AAS offers great ecological advantages over OPC at some levels such as the utilization of industrial by-products and overcoming the issue of disposal, low energy consumption, and low greenhouse gas emission. Apart from these environmental benefits, the novel binder has shown enhanced resistance to high temperatures and chemical attacks. However, many studies have mentioned the risk of its considerably higher drying shrinkage and early-age cracking compared to OPC concrete. Despite the abundant research on the self-healing mechanism of OPC, limited work has been devoted to studying the self-healing behavior of AAS. Self-healing AAS is a revolutionary product that provides the solution for these drawbacks. This study is a critical review of the self-healing ability of AAS and its effect on the mechanical properties of AAS mortars. Several self-healing approaches, applications, and challenges of each mechanism are taken into account and compared regarding their impacts.

## 1. Introduction

Infrastructures are always susceptible to severe changes in durability induced through multiple cracking. These changes are often related to the influence of shrinkage, mechanical loads, and/or environmental impact. Durability has a drastic influence on the structural resiliency and sustainability of concrete structures [1,2,3,4]. Once cracking occurs an additional pathway is provided for the transportation of moisture and aggressive agents such as calcium chloride, sulfate ions, gypsum, and carbon dioxide which could lead to alkali–aggregate reactions, reinforcement corrosion, and deterioration [5]. Consequently, restoring the inherent concrete properties is not an easy process. Traditional repair and maintenance including using mortar or epoxy gels cannot be considered cost-effective and environmentally friendly procedures. In some cases, the cost of restoration could be higher than that of the initial construction. For instance, by 2021, USD 1.6 trillion was needed for the maintenance and retrofitting of highway concrete bridges in the US [6]. Thus, the development of concrete material that has the capability of self-healing and regaining any loss of performance could a promising solution for these challenges.

Biological systems such as human organs and animal bodies are capable of self-healing and restoring their original functions. Scientists are inspired by this concept and realized the ability of concrete to heal cracks. Self-healing was discovered in cementitious material where concrete was found to heal cracks under normal conditions [7,8]. Neville [9] stated that tiny concrete cracks were completely closed under wet environmental conditions because of the re-initiation of the hydration reaction of the binding material. Edvardsen [10] pointed out that calcium carbonate is the main material observed within the crack width. Based on a literature review, two primary mechanisms exist for self-healing deteriorated concrete elements: autogenous and autonomic healing. Autogenous healing seals cracks naturally with no need for any external interventions through the precipitation of CaCO_3_ and/or hydrating the unreacted cement particles and producing extra C-S-H in the crack flanks. Meanwhile, autonomic healing is activated through means of artificial healing materials, for instance, bacteria, chemical admixtures, or microencapsulation [1,2,11,12].

In recent years, alkali-activated composites have been considered a strong competitor to replace ordinary Portland cement (OPC) concrete in various applications. The utilization of OPC is an energy-intensive process that consumes a large number of natural resources and releases a large amount of carbon dioxide gas. It was found that for the production of 1 ton of cement, 1.5 tons of raw materials is consumed and 1 ton of carbon dioxide gas (CO_2_) is emitted into the atmosphere [13,14]. It is estimated that 12% of the total CO_2_ emissions in 2020 were due to OPC production [15].

AAS is mainly derived from the chemical reaction between aluminosilicate solid precursors with a concentrated alkaline solution. The aluminosilicate precursor could be a beneficial industrial or mining by-product such as metallurgical slag from iron production or fly ash from coal combustion. This technology has a clear influence on reducing carbon dioxide emissions and innovating a sustainable environmentally friendly infrastructure. AAS has demonstrated unconventionally superior compressive strength and appropriate resistance to high temperature and chemical attack [16]. Nevertheless, many types of research have reported many limitations on using AAS in full-scale applications, including its higher drying shrinkage and early-age cracking compared to OPC concrete which threaten structural integrity, serviceability, and safety. This shortcoming has considerably impeded its broader construction applications [17,18,19,20,21,22]. Therefore, studying the ability of these innovative materials for crack self-healing has paramount importance.

## 2. Concrete with Slag Basis

Alkali-activated binders are experimentally proven to exhibit excellent mechanical properties among different types of binders. Nevertheless, there are still some obstacles encountered in their large-scale applications. Volumetric instability (i.e., massive shrinkage) is one of the most important challenges of alkali-activated concrete, especially with a slag basis. It was reported that the shrinkage may reach about two to four times higher when compared to OPC [17,23,24]. This significant shrinkage induces non-uniform deformation and causes tensile stress, causing matrix cracks. The induced cracks originate open space for penetration of various corrosive substances. Hence, a serious reduction in the load-bearing capacity, along with durability problems, occurs. Previous studies [25] proposed that the shrinkage properties are more complicated in slag than in OPC due to the complex hydration process. 

The higher shrinkage reported in AAS composites compared to OPC is associated with the finer pore structure and lower stiffness, as illustrated in Figure 1 [20]. In addition, there are larger numbers of mesopores (i.e., pore sizes between 2 nm, greater than micropores, and 50 nm, less than macropores) coupled with high capillary stress due to the pore distribution compared to OPC. The hydration reaction products (C-S-A-H and C-S-H) are also reported as a major cause of this phenomenon [7]. The presence of alkali metals in the AAS matrix (C-A-S-H gel) disrupts the regularity of the matrix layers’ arrangement, producing gels more prone to redistribution and exhibiting pronounced visco-elastic/visco-plastic characteristics [15,25].

Shrinkage takes place in two phases, plastic and hardening stages, and it is described as an inherent property of cementitious material. Slag paste and alkaline activators are the main materials in AAS concrete. Based on the different ways shrinkage takes place, it is classified into plastic, autogenous, drying, and carbonation shrinkages. Plastic (or young) shrinkage takes place immediately upon pouring concrete into a mold, and water evaporation is the main cause of this phenomenon. Autogenous shrinkage (chemical shrinkage) is an unavoidable reduction in volume produced by the macroscopic process of self-desiccation and the chemical reaction of concrete. Autogenous shrinkage commonly occurs when concrete has a low cement-to-water ratio [18,19]. Drying shrinkage is the change in volume due to moisture removal from the superficial gel pores that continue progressing along with the drying process of the paste component [19]. Carbonation shrinkage takes place because of the penetration of CO_2_ into the external surface of the binder [20].

Consequently, the predominant shrinkage mechanisms in alkali-activated composites are autogenous and drying shrinkage, which are linked to losing the internal hydration water and the formation of capillary stress [18] and the capillary suction pressure that occurs due to the very fine porosity of AAS [19]. It was reported that most AASs undergo higher drying and autogenous shrinkage than OPC-based material, which progresses the cracking [26]. Autogenous shrinkage is more independent of the alkaline dosage and surroundings. The proportion of autogenous shrinkage was reported to be considerably steady compared to drying shrinkage that increased gradually with the increase in alkaline solution. The increase in the alkaline solution is represented in terms of silicate modulus and alkaline dosage, see Figure 2 [24]. However, another study [17] has highlighted that autogenous and drying shrinkage take place simultaneously under sealed conditions. Accordingly, it is not easy to determine each of them separately. The detected drying shrinking often incorporates a part of the autogenous shrinkage, however, the drying shrinkage of slag-based matrix is 1.6 times that of concrete when using traditional measures, with unrestrained shrinkage shown by 75 × 75 × 285 mm^3^ prisms as per Australian Standard AS1012.13 at 23 °C for 24 h followed by exposure to 23 °C and 50% relative humidity (RH). Thus, the only way to differentiate between drying and autogenous shrinkage is to use the developed ring test or the generation of restrained shrinkage prisms by modifying the Australian Standard AS1012.13 [27,28,29,30]. Another way of testing restrained shrinkage is using the ellipse ring as suggested by Kheradmand et al. [28] when studying the performance of fly ash alkali-activated mortars.

There are several ways mentioned in the literature to mitigate the high shrinkage in AAS. These ways include factors that might not only influence the shrinkage but might adversely affect mechanical and fresh properties of the produced concrete. Table 1 discusses these factors that might influence the shrinkage in terms of durability such as alkaline activator type, activator concentration and its ratio to the binder, mineral admixture, chemical additive, nanoparticles, curing condition, and fiber reinforcement, which could be used to mitigate the high shrinkage in AAS pastes.

As is clear from Table 1, the alkaline activator was evaluated by many studies on the shrinkage and mechanical properties of AAS. Two factors appeared to affect the alkaline activator: type and concentration. Figure 2 illustrates that the increase in sodium silicates (0.5 M to 2 M) results in higher expansion percentage. Similarly, Figure 3 confirms that higher alkali dosage (4%, 6%, and 8%) leads to higher autogenous and drying shrinkage. The reduction of alkaline solution concentration lowers the percentage of medium capillaries and pores, which in turn increases large capillaries that reduce the capillary tensile stress.

On the other hand, replacing the slag with mineral admixtures such as silica fume and fly ash would weaken the strength development. Several researchers [21,22] explored the use of various mineral admixtures: fly ash, silica fume, metakaolin, and lime. Seo et al. [37] reported a shrinkage improvement when incorporating lime as a partial replacement for slag due to the physical filling enhancing the paste’s compactness. However, the compressive strength results were reduced by 66% due to the formation of crystal products of Ca(OH)_2_ [37]. Moreover, metakaolin has been recorded to effectively alleviate the autogenous shrinkage of slag pastes. This reduction was illustrated by the existence of extra Al and Si in metakaolin, which diminished the concentration of Na^+^ and OH^−^, retarding the hydration reaction. Although there was slight variation in the studies that explored mineral admixtures’ influence on shrinkage and strength, many scholars suggested that, generally, the mineral admixtures can help in reducing shrinkage but lead to a reduction in the strength of the AAS. A few suggested researchers [7,26] that mineral admixtures’ incorporation does not affect the shrinkage. Zhang et al. [7] stressed that selecting a suitable mineral admixture proportion is of high importance to optimize the performance of AAS.

Similar to the mineral admixtures, chemical admixtures could influence the shrinkage, especially regarding the expansion and shrinkage of concrete using shrinkage-reducing admixtures, expansive agents, and superabsorbent polymers. The idea mainly relies on reducing the surface tension of the pores by increasing the coarseness of the pore structure and increasing the micropores [7,26]. Nevertheless, other researchers [20] suggested that the chemical admixture might have a negligible effect on reducing the shrinkage of AAS and this needs to be explored intensively.

Furthermore, fibers are another method that effectively controls shrinkage. The inclusion of fibers reduces shrinkage and enhances flexural strength through ties that resist tensile stresses resulting from pore capillaries. Fibers serve as a restraint in the composite, known as the fiber-bridging effect, creating composite stability and resisting any change in the paste volume. However, it should be highlighted that optimization of fiber dosage is an important aspect as it could negatively influence both shrinkage and strength. Puertas et al. [38] and Hammad et al. [13] recommended the optimized fiber range of 0.5 to 1.0%; however, this range differs based on the fiber type, silicate modulus, alkali dosage of the AAS mix, and curing condition.

As curing affects the addition of fiber, it is a factor that affects the shrinkage. Shrinkage is reduced when elevated-temperature curing occurs as reported by Awoyera and Adesina [26]. Their results showed that early elevated temperature transforms the silica gel into calcium silicate hydrates. This action improves the adhesion property found within C-A-S-H gels, accelerating the strength gain. Moreover, the overflowing products from the hydration process are redistributed and surrounded by the unreacted binder, resulting in a coarse pore structure [26]. Collin and Sanjayan [20] added that bath curing of AAS with a shrinkage admixture would create a smaller crack width than that of OPC at 3 and 7 days of curing, which indicates that usually curing has accompanying factors that affect the crack width reduction.

All these mitigation processes reduce the shrinkage up to a specific threshold; however, these processes might adversely affect the overall compressive and flexural strength or increase the embodied energy in AAS systems [26]. Furthermore, other mitigation processes involving aggregate types could be another way of reducing the shrinkage. Collin and Sanjayan [29] reported that AAS beams with slag aggregate performed better than those with natural aggregate in terms of shrinkage and tensile strength; however, these beams provided a lower elastic modulus than those of natural aggregate. Hence, a bio-green alternative could be used to reduce shrinkage and enhance compressive strength as well.

Not only would shrinkage cracking be the main issue to overcome in AAS but also its high susceptibility to cracking because of the intrinsic brittleness [35,36,37]. Generally, cracking influences the safety, serviceability, and integrity of AAS structural elements, which limits their commercial application. Moreover, Zhang et al. [39] reported that conventional maintenance techniques are typically laborious, onerous, and time- and money-consuming [3]. Thus, there is a need to develop a sustainable AAS application using the recently developed techniques of self-healing AAS, occurring autonomously without the intervention of humans, facilitating the repairing of cracks and the creation of resilient AAS. This paper reviews the most important self-healing techniques used for conventional concrete generally and applied to AAS.

## 3. Self-Healing Mechanisms

Many researchers [1,2,4,11,40,41] discussed the selection of several techniques for self-healing including the two main types, autogenic and autonomic, based on the techniques’ availability and the characteristics of cracks in terms of width and depth, as illustrated in Table 2. The values of crack width and depth listed in Table 2 were proposed by multiple researchers based on their own experience with specific materials, concrete mix, curing conditions, and experimental procedures.

Cracks, on either a macro or micro scale, could significantly induce a series of problems in terms of deflection and aesthetics, interfering with durability and serviceability. Consequently, crack characterization of structural elements is controlled by the serviceability limit state based on the code and guidelines required for design. The crack width and depth should be the main aspect of self-healing technology; however, as shown, self-healing could provide a reduction in crack width of nearly 1 mm (≈970 µm) which exceeds the size of most of the limited cracks under serviceability loads as per ACI 318. Thus, several techniques and mechanisms are reviewed in the next section to understand how these mechanisms and techniques would work in healing the cracks of AAS structures.

### 3.1. Autogenous Self-Healing

Autogenous healing is the ability of concrete or the conventional ingredients of the cementitious matrix to recover from damage. The specific chemical composition of the cementitious matrix is the main reason for this process to take place [51]. Generally, about 20 to 30% of cement in conventional concrete remains unhydrated during the process of strength gain [4,10,11]. These unhydrated cement particles react with water once cracking initiates. Thereby, the hydration process starts again, producing hydration products such as CaCO_3_ to seal/fill the cracks [4,10,11]. 

According to substantial experiments and practical application, an autogenous self-healing mechanism is a combination of complex chemical and physical processes. As stated by most of the studies in the literature [4,10,11], various causes could be responsible for the self-healing of cracks, such as (a) formation of calcium hydroxide or calcium carbonate; (b) the blocking of cracks by impurities in water; (c) hydration of unreacted cementitious materials; and (d) swelling of the hydrated cementitious matrix (C-S-H) in the crack flanks [2,4].

More than one of the proposed causes can occur simultaneously. However, most of the mechanisms can partially fill the entrance of some cracks but cannot fill the crack. This could help in preventing the development of further cracks or control the deep penetration of harmful chemicals. Crystallization of calcium carbonate is the most effective method to heal the crack naturally. This is proved by the common observation of macroscopically and microscopically precipitated calcium carbonate on the crack surfaces as a white residue. Dissolving one of the cement hydration products in water is an essential condition for calcium hydroxide to be liberated and dissipated along the cracking surfaces [4,10,11]. This means that this reaction mainly depends on the presence of water [52]. In this way, the calcium ions produced from the hydration process become able to react with the dissolved carbon dioxide, then self-healed crystals are formed and fill the gaps as described in the following Equations (1)–(5) [4,10].
H_2_O + CO_2_ ↔ H_2_CO_3_
(1)
(2)$$H_{2}{CO}_{3}↔H^{+}+{HCO}_{3}^{-}$$
(3)$${HCO}_{3}^{-}↔2H^{+}+{CO}_{3}^{2-}$$
(4)$${Ca}^{2+}+{CO}_{3}^{2-}↔Ca{CO}_{3}\;(pH\;water>8)$$
(5)$${\mathrm{Ca}}^{2+}+{HCO}_{3}^{-}↔CaCO_{3}+H^{+}\;(7.5<pH\;water<8)$$

Locations of CaCO_3_ crystals depend on the pH, the surrounding temperature, the partial pressure of CO_2_, the calcite saturation, and the concentration of Ca^2+^ and ${CO}_{3}^{2-}$ in the solution [10]. The other proposed mechanism of self-healing caused by further hydration of unhydrated cementitious components was recently discovered to be applicable only at the early age of concrete as the formation of calcium carbonate is the most likely reason for self-healing at later ages [9].

By observing surface cracks, calcium carbonate and calcium silicate hydrates were recognized as the main products of the self-healing process. The percentage of the formed C-S-H (<15%) was considerably lower than the percentage of the produced calcite which was 80%. It was realized that there is an obvious decline in the intensity of the self-healing efficiency after about 300 h [49]. Autogenous self-healing takes place through three processes: physical, chemical, and mechanical. The physical process causes the matrix to swell. The chemical process is responsible for initiating the ongoing hydration of the unreacted particles to precipitate CaCO_3_. The mechanical process comprises crack closing with the generated particles from the crushed concrete surface or impurities from water ingress [52]. 

Different studies were conducted to study the efficiency of the autogenous self-healing mechanism and the factors affecting it. It was found that this process is effective for small crack sizes, however, sometimes larger crack widths (between 200 µm and 300 µm) were observed [2]. Based on previous observations, the following factors (listed in Table 3) considerably affect the autogenous self-healing process [2,4,12,41,51,52].

Limited research has been devoted to studying the autogenous self-healing property of AAS [15,53,54]. The crack healing in AAS mortar specimens was inconspicuous in most of these studies, especially when the initial crack width increased from 50 µm to 100 µm, followed by a gradual decrease in any further increase in the crack width [53]. This could be explained by understanding the properties of AAS. AAS systems are commonly known for their early high strength. Hammad et al. [13] have proved that AAS composites gained about 85% of their 28th day compressive strength after only 1 day of casting. The percentage of strength gain increased to 98% and 99.5% after 7 and 14 days. The high pozzolanic activity of AAS systems and the presence of high calcium and silicon content create a reaction much faster than the reaction that occurs within OPC concrete because of the availability of highly alkaline conditions as well as the greater fineness of AAS [13,55,56]. Consequently, most of the calcium content is consumed in the gepolymerization process during the first 7 days. Some modifications have been proposed in AAS mixtures to improve the autogenous self-healing property such as partial replacement of AAS by calcium hydroxide [53,57], addition of fiber [53,54,58,59], and incorporation of an additional calcium source (i.e., calcium lactate) [15,54]. A study conducted by Provis showed that incorporating Ca(OH)_2_ in the AAS mix led to higher Ca^2+^ concentration due to the low solubility of Ca(OH)_2_ at a high pH level before cracking takes place. Consequently, Ca^2+^ was not consumed during the geopolymerization process. After the activation process of AAS, Ca(OH)_2_ dissolved in the curing water, providing calcium carbonate precipitation [53,60] as illustrated in Equations (6) and (7).

The addition of polyvinyl alcohol (PVA) and polyethylene (PE) fibers showed a tendency towards enhancing the healing property of AAS [53,54,57,58,59]. SEM images of AAS specimens incorporating PVA fibers revealed many crystals covering the crack surface, highlighting that PVA fibers promoted the precipitation of healing products of CaCO_3_ through various intermolecular forces such as ionic bonding, hydrogen bonding, and van der Waals forces [15]. Another study conducted by Nguyễn et al. [59] clarified that the addition of PE fibers bridged cracks and enhanced the precipitation of healing materials by providing nucleation sites. They stated that the healing product crystals were not only found on the external surface of the crack but also formed on the fibers. Zhang et al. [54] tested the addition of calcium lactate and urea in an AAS mix incorporating PVA fibers. The findings showed a slight improvement in crack surface healing.
(6)$$2{\mathrm{OH}}^{-}+H_{2}CO_{3}↔{CO}_{3}^{2-}+2H_{2}O$$
(7)$${Ca}^{2+}+{CO}_{3}^{2-}\to CaCO_{3}\;(pH\;water>8)$$

### 3.2. Autonomous Self-Healing

Recently, autonomous self-healing has attracted considerable attention. Autonomous concrete healing is an artificial healing mechanism that occurs when an appropriate healing agent is added to concrete. Accordingly, the healing and blocking of cracks can be carried out automatically without any external aid or repair at ambient room temperature. This type of healing takes place through chemical and biological processes [2,44,45,52].

#### 3.2.1. Chemical Self-Healing Process

One of the new approaches in autonomous self-healing uses the injection of chemical compounds into concrete cracks. This chemical process could be carried out by mixing fresh concrete with (liquid) chemical reagents such as glue. For accomplishing this process, there are several methods, for example, hollow pipette and vessel networks containing glue and encapsulated glue. 

##### Hollow Pipette and Vessel Networks Containing Glue

The hollow pipette concept is inspired by blood vessels in animals. These pipettes or vessels (classified by diameter size) provide a medium for storing functional components or healing agents. Pipettes are embedded within the composite matrix. When concrete is subjected to damage or cracking occurs, the healing material will flow out to start the healing process. Either the hollow pipettes or the vessel networks can be used to design a self-healing concrete based on passive or active mode. The passive mode uses hollow vessels that are not linked to an external glue source, while the active mode needs an external supply of glue for distribution [11]. Systems of hollow vessels have been investigated at different length scales through several studies on the design of different healing materials such as polymers and polymeric composites. The applicability of hollow vessels in allowing the flow of healing agents was proven in many cases [4]. This method has been applied to cementitious materials and demonstrated as a feasible approach by various researchers. Experiments have been carried out on concrete specimens containing distributed brittle vessels of healing materials inside. Then, the pattern of glue distribution within cracks was monitored upon concrete damage. For accurate monitoring, vessels were sometimes filled with healing agents mixed with fluorescent dye for detecting the rupture details and sequence [61]. 

Methyl methacrylate, ethyl cranoacrylate, acrylic resin, and epoxy are different examples of healing materials (glue) that are suitable for filling the hollow pipettes. A network of vessels was applied within a concrete specimen for the distribution of glue. This brittle vessel network inside concrete was connected at one end to a glue supply, whereas the other end was sealed inside concrete. Another study proved that this technique used in a concrete specimen resulted in a 20% increase in the loading capacity under a subsequent flexural test [11]. Despite there having been several studies on using this procedure, there are no available data on applying this mechanism in AAS concrete. Hence, this method is considered a complicated method that is subject to a lack of constructability using the current technology. Therefore, it needs verification for use in actual projects [61,62].

##### Encapsulated Glue

The encapsulation technique was inspired by examples in nature, ranging from the macro scale (bird’s egg or seeds) to the micro scale (cells). Microencapsulation involves the development and preparation of capsules containing dyes that were originally for paper copying and eventually replaced with carbon paper. White et al. [63] developed the idea and introduced the applicability of using microcapsules containing glue for designing self-healing concrete. Concrete cracks would cause a rupture in the embedded microcapsules, causing release of the glue into the crack faces through capillary action [1,11].

Many researchers believed that this technique is versatile due to its efficiency in regaining the mechanical properties and the durability of concrete [2,64]. This method was initially applied to structural polymers [63]. The proposed capsules contain the healing agent and a catalyst for chemical triggering. There are three main components in the mechanism of capsule-based self-healing: (a) the trigger that initiates the process of activation by damaging the capsule, (b) the healing agent which is the core that is released upon activation into the crack, and (c) the capsule shell (coating for protection) to prevent the direct contact of the healing agent and the surrounding matrix. The compatibility of these materials with the concrete matrix is a crucial factor for a successful self-healing process [52]. There are two mechanisms for activating the capsules. The first and the most popular procedure is the mechanical activator where the stresses resulting from the crack formation lead to the breakage of the brittle capsule and releasing of the agent. The second mechanism depends on adding chemical triggers for penetrating the matrix via nanocracks before microcracks occur. The sensitivity of the chemical trigger was deduced to be higher [65].

Different types of healing agents are used for the self-healing process as listed in Table 4. They are categorized into three groups: one, two, and multiple healing agent components. There is no need for more chemical components or catalysts (such as sodium silica (Na_2_SiO_3_)) in the case of using one component as a healing agent for activation due to its ability to act individually. For the other two types, another substance, for example, dicyclopentadiene (DCPD), is required to achieve maximum efficiency. Practically, two or more component systems are found to be more complicated and have a high risk of inappropriate mixing that may cause inefficient crack healing. Moreover, optimum fluid with low viscosity is a vital factor for controlling the healing material transportation and penetration through the crack as well as solidifying in the desired locations [66]. Basically, after conducting many studies using this approach, it was elaborated that there were many technical problems, including: (a) mixing two healing components is difficult, (b) the required quantity of agent for filling one microcapsule is limited, (c) the capsule shell should be strong enough to protect the healing agent during the mixing and hydration process (different types of applicable shells are illustrated in Table 5), (d) the bond between the capsule shell and the concrete matrix has to be improved and be stronger than the microcapsules [11,64].

Despite the remarkable progress accomplished in adopting this technique in OPC-based systems, no studies have introduced microcapsules in AAS-based composites and evaluated their performance.

#### 3.2.2. Biological Self-Healing Process

This process is an environment-friendly mechanism that takes place by adding microorganisms to concrete. For that, it has been categorized as a biological strategy. The reason for choosing microorganisms is their wide availability (i.e., in soil, water, acidic springs, industrial wastewater, oil reservoirs, etc.). There are several types of microorganisms, for instance, bacteria, fungi, and viruses. Among these types, it was found that a certain strain of bacteria could be considered due to its ability to precipitate certain useful chemicals used in the design of biological self-healing concrete. These useful precipitation chemicals are polymorphic iron aluminum silicate ((Fe_5_Al_3_)(SiAl)O_10_(OH)_5_) and calcium carbonate (CaCO_3_) [40,41]. 

For governing bacterial self-healing, there are two processes: (a) bacterial metabolism, and (b) enzymatic ureolysis. In the first mechanism, bacteria are present as a catalyst that converts the precursor compound into a filler material. The produced filler material is a calcium carbonate-based mineral that acts as a type of bio-cement and seals cracks [77]. The reaction taking place can be formulated as follows:Ca(C_3_H_5_O_2_)_2_ + 7O_2_ → CaCO_3_ + 5CO_2_ + 5H_2_O(8)

This type of reaction is six times more effective compared to autogenous self-healing as the produced CO_2_ starts a localized reaction with Ca(OH)_2_ within the crack mouth. As a result, five more CaCO_3_ molecules are produced [78]. 

The second process is known as ureolysis which is one of the most popular ways to design self-healing concrete. In this process, bacteria can produce urease, an enzyme that catalyzes urea (CO(NH_2_)_2_) into ammonium (${NH}_{4}^{+})$ and carbonate (${CO}_{3}^{2-}$). The hydrolyzed ammonia and carbon dioxide help in increasing the pH and carbonate concentration in the presence of a bacterial environment. A series of biochemical reactions occur to produce calcium carbonate as described in the following Equations (9)–(15). The primary role of bacteria is to form the precipitate calcium carbonate which is attributed to their effect in elevating the pH of the environment through their metabolism [11,79,80]:

Urea is hydrolyzed into carbonate and ammonia in the presence of urease as shown in Equation (9):CO(NH_2_)_2_ + H_2_O → NH_2_COOH + NH_3_(9)

Carbonate is simultaneously hydrolyzed to produce ammonia and carbonic acid: NH_2_COOH + H_2_O → NH_3_ + H_2_CO_3_(10)

Carbonic acid is hydrolyzed into carbonate ions and hydrogen ions:(11)$$H_{2}CO_{3}↔{HCO}_{3}^{-}\;+\;H^{+}$$

Ammonia is spontaneously hydrolyzed to produce ammonium and hydroxide ions:(12)$$2{\mathrm{NH}}_{3}+2H_{2}O↔2{NH}_{4}^{+}+2{OH}^{-}$$

The production of hydroxide ions increases the pH value and causes the overall equilibrium to form carbonate ions:(13)$${HCO}_{3}^{-}+H^{+}+2{NH}_{4}^{+}\;+\;2{OH}^{-}↔{CO}_{3}^{2-}+2{NH}_{4}^{+}+2H_{2}O$$

The negative charge of the produced carbonate ions attracts the positively charged calcium ions (Ca^2+^) to form the precipitation of calcium carbonate (CaCO_3_) at the cell surface:Ca^2+^ + Cell → Cell-Ca^2+^(14)
(15)$$\mathrm{Cell}\text{-}{\mathrm{Ca}}^{2+}+{CO}_{3}^{2-}\to Cell\text{-}CaCO_{3}↓$$

A large number of bacteria have been tested to produce self-healing concrete, however, the survival of most of them is in still in question due to the high pH of the fresh concrete (between (10) and (13)). This high-alkalinity solution is considered to be harsh for the bacteria’s survival. Furthermore, the fresh concrete’s temperature, which is around 70 °C, is too high for cell growth. Not only is temperature one of the drawbacks but so is water scarcity for cell growth once fresh concrete hardens. As a result, the chosen type of bacteria has to show high resistance against the high pH, temperature, and massive limitation of water [81]. The bacteria must have the ability to withstand the pressure caused by the hydration and densification of the mix. Over time, the pores in the matrix will reduce due to the ongoing hydration that leads to exposing the bacteria to considerable compressive stress. Consequently, the chosen bacterial strain must be an obligate alkaliphile as well as spore-forming. These strains are fully functional vegetative cells that become dormant and produce spores under adverse conditions. Spores are much tougher than vegetative cells and can resist harsh conditions. Once the environmental conditions improve, these spores return to their vegetative state. Hence, these spores remain dormant until cracks start and, once exposed to water, they will germinate back into vegetative cells and metabolize the available growth substrate, producing the precipitation of healing products [39]. Bacillus species are specifically highlighted to have the potential for application in cement materials and are known for their ability to precipitate CaCO_3_ in different ways, closing pores and cracks in concrete and cementitious materials [12,46,77,81,82]. 

The effects of bacteria on the compressive strength and transport properties of concrete are listed in Table 6. Nevertheless, the variation in concrete porosity during maturing may cause a decrease in the viability of spores as well as their survival time which may only reach 4 months [83]. Consequently, so-called protection solutions and organic precursors have been introduced [41,77,84]. Finally, the process of microbial-induced calcite precipitation (MICP) can be used in a cementitious matrix composed of three main elements: (a) an alkali-resisting, spore-forming, Gram-positive bacterium, in an inactive form, which can be activated upon exposure to water, (b) a nutrient or precursor compound, and (c) a carrier compound for protection of bacteria from the external pressure applied by concrete [85]. However, the incorporation of bacteria is not limited to the spore form only. Bacteria could be added by encapsulation or through using the vascular network method [2,11]. However, the encapsulation technique and vascular network method are difficult to prepare in small or large amounts which makes the processes not cost-effective choices. Moreover, the spores of bacteria cannot break the encapsulation shell easily until they encounter major cracks or deterioration. Additionally, the spore form of bacteria shows the fastest healing reaction [86]. The main advantage of this mechanism is that it is a natural, environmentally friendly method and compatible with the cement matrix. Its main disadvantage is the required measures for the protection of bacteria [87]. A summarized comparison between different self-healing mechanisms is provided in Table 7.

## 4. Self-Healing Alkali-Activated Concrete

According to the literature, few studies have investigated the self-healing property of alkali-activated concrete. Zhang et al. [92] explored the potential of self-healing in AAS mortars using PVA fibers and compared it with the behavior of cement-based materials. Based on optical microscopy and scanning electron microscopy (SEM), both composites exhibited self-healing properties, producing calcium carbonate as the main healing product. Another study showed that the addition of PVA fibers within AAS mortar specimens achieved a maximum healing ratio of 65% for the narrowest crack width of 50 μm and the lowest healing ratio of 18% for the original crack width of 150 μm. This efficiency could be improved by incorporating crystalline additives to reach 100% and 25% for crack widths of 50 μm and 150 μm, respectively [97]. Similarly, AAS composites incorporating PE fibers were revealed to have a superior reduction in crack width concerning cement-based composites. SEM observations have confirmed that the prominent healing material of both OPC and AAS-based composites is CaCO_3_ as previously stated by different researchers. Moreover, the tensile strength along with the tensile strain increased significantly by 150% and 234% during the reloading tests after crack healing [57,59]. The effect of the partial replacement of slag by calcium hydroxide on the healing process has shown higher levels of crack healing compared with specimens without calcium hydroxide because of the availability of higher concentrations of Ca^2*+*^ ions in the AAS matrix. This led to reducing the diameters of capillary pores by about 37% and 23% in calcium hydroxide and zero calcium hydroxide specimens [53]. The potential of the bacterium *Sporosarcina pasteuri* in the crack healing property of AAS mortar was investigated using porous expanded recycled glass granules as bacterial carriers. Most of the crack surfaces under 100 μm were entirely healed after 5 months of curing. Bio-based samples showed enhanced crack healing with a maximum width of 140 μm. Increases in compressive strength of 7% and 14% have been observed for bacterial-based specimens incorporating calcium lactate and calcium chloride, respectively. The usage of a combination of bacterial nutrients (calcium lactate and urea) and porous expanded recycled glass granules led to a decrease in mechanical properties. However, the final strength increased by 7% due to the positive effect of the immobilized bacteria [15,54]. Consequently, the incorporation of bacteria in AAS seemed to be a promising technology for widening full-scale production and overcoming the problem of early cracking in AAS systems [15,53,54,98].

## 5. Conclusions

The innovative technology of alkali-activated composites is known for its small carbon footprint and energy savings. These smart sustainable composites offer an eye-catching combination of good-quality mechanical properties and durability. However, degradation initiated at a very early stage may put a limitation on their usage as construction materials. The volumetric instability of AAS is two to four times higher than that of OPC composites. The finer pore structure of AAS, the lower stiffness, and the presence of alkali metals in the hydration products (C-S-A-H) are reported to be the main causes of this higher shrinkage. Based on this critical review, this research suggests producing smart alkali-activated composites with self-healing properties, which would provide a self-repairing structure with the minimal help of external maintenance.

The natural self-healing mechanism could occur due to any of the following reasons: formation of calcium hydroxide or calcium carbonate, further hydration of the unreacted particles of cement, swelling of the hydrated cementitious matrix (C-S-H) in the crack flanks, or blocking of cracks by impurities in the curing water. More than one of the mentioned causes may occur.Clinker content in cement, hydration age, crack geometry, exposure condition, and addition of fiber or additives could enhance the efficiency of autogenous crack healing.Innovative techniques of chemical and biological self-healing have been proposed. The mentioned strategies are hollow vessels, encapsulation, and immobilization of bacteria. The idea of these methods is to prompt the intrinsic self-healing property of concrete to produce calcium ions and supply the cementitious matrix with external chemical or functional materials to provide better healing capacity.Bacterial-based bio-concrete is a remarkable solution because it showed enhanced results and an eco-friendly practical process. Nevertheless, the efficiency of the healing process depends on the proportion of the unreacted matrix that will respond to the introduced healing agent, and crack geometry and size play a vital role in predicting effective healing.From the available limited literature, it seems feasible to prompt the production of alkali-activated self-healing composites with a good crack-healing ratio and improved mechanical properties.

## 6. Future Research and Recommendation

As discussed earlier, several parameters could influence the shrinkage of AAS and there are several solutions to overcome this shrinkage. The most commonly suggested is an autonomic healing solution. However, very few researchers work in this area, and the following points still need to be discussed:Curing methods, including cycling air and wet methods, wetting and drying for just 24 h, wetting and air drying for 3 h, air drying, and oven drying. To date it is not clear which curing method would be suitable.Although it was stated that sodium concentration is a parameter, limited studies have mentioned the possibility for bacteria to die before activation; however, none of them mentioned the concentration percentage at which the bacteria will not be influenced.

## Figures and Tables

**Figure 1 materials-16-04394-f001:**
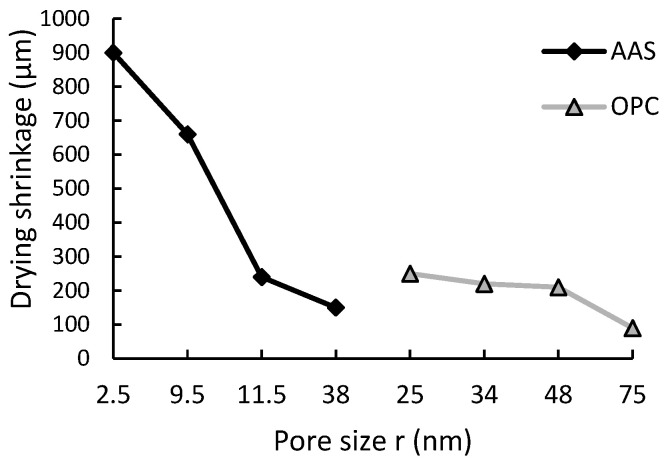
Drying shrinkage versus pore size for OPC and AAS paste [20].

**Figure 2 materials-16-04394-f002:**
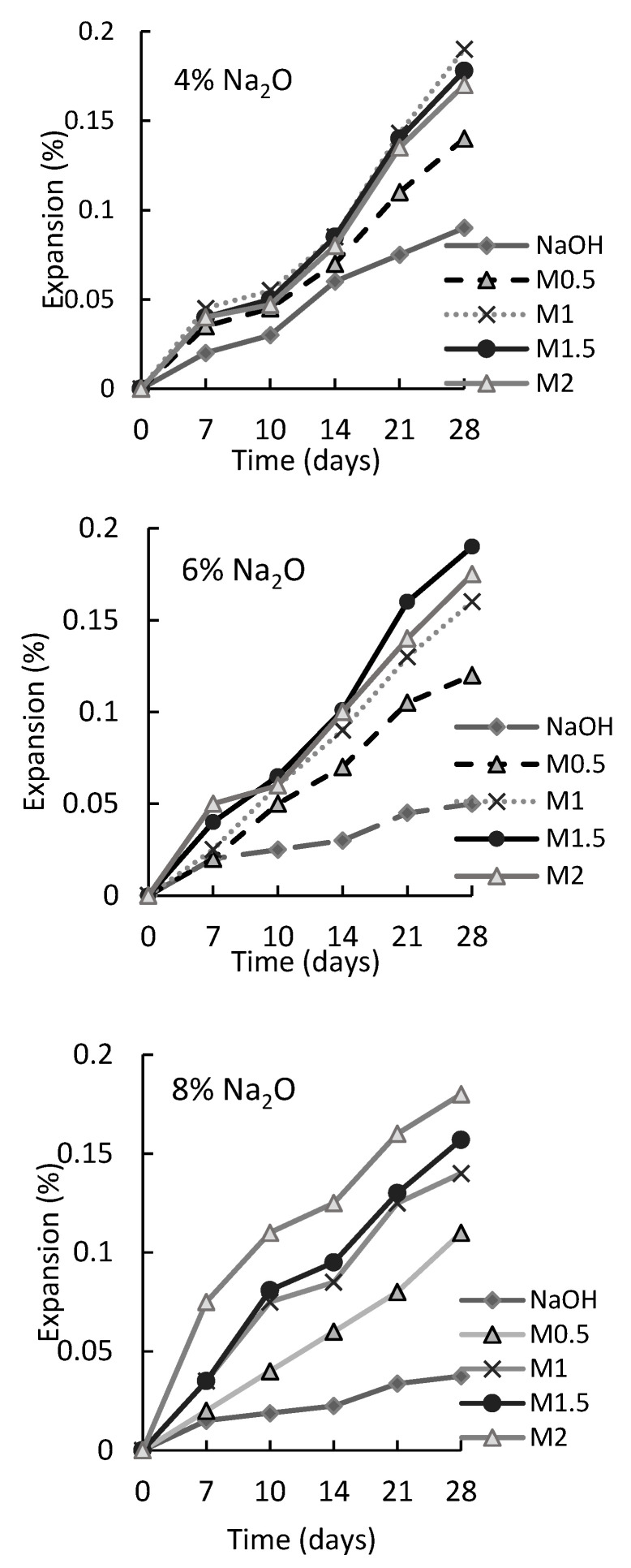
Effect of different alkali dosages and silicate modulus on the expansion of the AAS mortars [24].

**Figure 3 materials-16-04394-f003:**
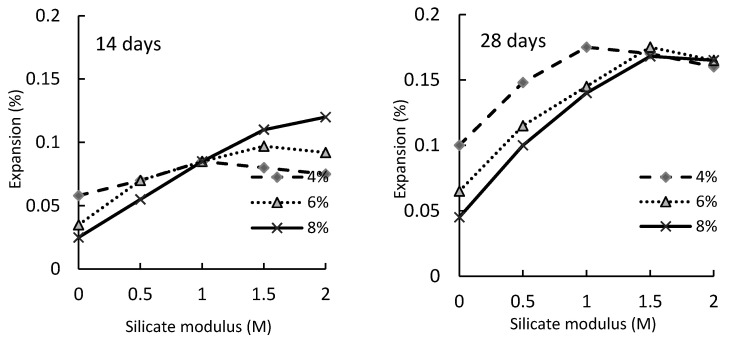
AAS expansion at different alkali dosages and silicate moduli after 14 and 28 days [24].

**Table 1 materials-16-04394-t001:** The factors influencing the durability (i.e., shrinkage) and mechanical behavior of AAS.

	Factors	Action	Mechanical Properties	Durability (Shrinkage)	References
Alkali activator	Alkaline activator type	Decreasing the alkali activator modulus alkaline content		Reduces the autogenous and drying shrinkage	[31,32,33]
		Increasing alkali dosage within the practical range of 2–8% by slag silicate modulus	High compressive strength		[24,26]
	Concentration	Reduces the concentration	Reduces the compressive and tensile strengths	Lowers the proportion of medium capillaries and pores Increases large capillaries that reduce the capillary tensile stress	[7,26]
Mineral Admixtures	Partial replacement of slag by: lime	Fly ash	Weakens the strength development Decreases strength	Reduces the shrinkage of AAS	[34,35]
		Silica fume			
		Metakaolin	Slightly decreases the compressive strength	Improves shrinkage	[36]
		Lime	Reduction of 66% in the compressive strength	Alleviates shrinkage	[37]
Chemical admixture	Chemical additives such as shrinkage-reducing admixture, expansive agents, and superabsorbent polymers			Enhances the AAS shrinkage	[7,26]
Fiber reinforcement	The appropriate fiber range is 0.5–1.0%		Enhances the flexural and compressive strengths	Reduces shrinkage	[13,16,38]
Curing conditions	Elevated temperature curing			Decreases the shrinkage of AAS systems	[26]

**Table 2 materials-16-04394-t002:** Applicability of self-healing strategy with crack width and depth for OPC concrete.

Technique	Crack Width (µm)	Crack Depth (mm)	Reference
Autogenous healing	6	-	[42,43]
Bacteria	970	27.5	[44,45,46]
Encapsulation	970	35	[47,48]
Supplementary cementitious materials	200	-	[41,42,43,44,45,46,47,48,49]
Other chemical and biological methods	220	-	[50]
Polymer	0.138	-	[3]

**Table 3 materials-16-04394-t003:** Factors affecting the efficiency of autogenous self-healing mechanism and their influences.

Factor	Influence
Chemical composition of the binding material	The ratio of Ca/Si indicates the intensity of producing C-S-H or CaCO_3_ as healing materials; however, the content of calcium is more crucial for precipitating calcium carbonate, considered the main healing product.
Concrete age	Autogenous self-healing is a time-dependent process. The early age of concrete is more important due to the presence of more unhydrated bonding particles and its higher ability to form new C-S-H gel.
Crack shape and size	Geometrically, crack dimensions (width, length, and depth), as well as a crack pattern (branched or accumulated), indicate the extent of autogenous healing. The wider the crack (>200 µm), the easier the access for water and carbon dioxide. The overall closure of narrower cracks (around 50 µm) tends to be higher due to the easier filling.
Effect of exposure	Exposure to water is essential for initiating the chemical reaction and acting as a carrier medium for particles. Various water regimes induce different healing efficiencies. Water submergence appears to be the most effective exposure regime due to its higher ability for provoking the carbonation process and precipitating the CaCO_3_ compound. Nevertheless, a few studies have revealed that the wet and dry cycle works perfectly as compared to water submergence for self-healing because of the massive availability of carbon dioxide in the air.
Temperature	Elevated temperatures were reported to have a positive effect on the processes of crack closure as the chemical reaction of the self-healing process is greatly stimulated by temperature.
Effect of fiber	The influence of fibers on the healing mechanism is still not completely understood. Fibers could help in initiating the mechanism of self-healing. This assumption was based on spotting CaCO_3_ and C-S-H in mixes containing fibers during the healing process. This could be explained by the effect of fiber polarity on the chemical reaction. Polyvinyl alcohol (PVA) is known for its higher polarity effect compared to other fiber types. Polarity strength is defined as the existence of OH^−^ radicals that act as nucleation locations attracting calcium ions.
Effect of additives	Mineral additives have a positive effect on the autogenous healing process. Studies revealed the better performance of the healing process with the presence of crystalline additive or expansive additive within the concrete matrix. Additives directly enhance the material porosity, leading to the concentration of the precipitated calcium carbonate on the tip of the crack. Additionally, they improve the specimens’ pH for more precipitation of calcium carbonate.

**Table 4 materials-16-04394-t004:** Different types of healing agents used in the encapsulation technique.

Encapsulated Healing Agent	Reference
Sodium silicate	[66,67,68,69]
Epoxy	[40,64]
Ca(NO_3_)^2^	[70,71,72]
Dicyclopentadiene	[66]
Bacterial spores	[73]
Methyl methacrylate	[43]
Calcium sulfoaluminate	[74]
Silica solution	[75]
Minerals and expansive powders	[76]

**Table 5 materials-16-04394-t005:** Different types of capsule shells used in the encapsulation technique.

Capsule Shell	Reference
Polyurethane/urea formaldehyde	[66,67,68,69]
Silica	[43]
Glass	[41,76]
Ceramic	[41]
Polystyrene resin	[47]
Polyvinyl alcohol	[74]

**Table 6 materials-16-04394-t006:** The influence of various bacterial types on the compressive strength and transport properties of concrete.

Bacteria	Base Binder	Compressive Strength	Permeability	Water Absorption	Chloride Penetration Resistance	References
*Sporosarcina pasteuri*	Slag, fly ash, and cement	↑	↓	↓	↑	[88,89,90]
*Bacillus sphaericus*	Cement	↑	↓	↓	↑	[46,91]
*Bacillus subtilis*	Slag and cement	↑	↓	↓	↑	[92,93]
*Bacillus megaterium*	Cement	↑	-	-	-	[3,93]
*Bacillus cohnii*	Fly ash and cement	↑	-	↓	-	[89,94]
*Bacillus aerius*	Rice husk ash	↑	↓	↓	↑	[95]
*Bacillus pseudofirmus*	Blended cement	↓	-	-	-	[77]
*Diaphorobacter nitroreductase*	Cement	↓	-	-	-	[96]

**Table 7 materials-16-04394-t007:** The advantages and disadvantages of different self-healing mechanisms.

Self-Healing Mechanism	Advantage	Disadvantage
Autogenous	Natural, environmentally friendly method Mechanical property improvement Easy to implement	Microcracks (limited crack size) can be healed Its efficiency depends on the availability of unreacted cement particles A higher amount of cement could lead to a matrix more susceptible to shrinkage and cracking An increase in CO_2_ emission because of the higher amount of cement
Active (external) hollow vessels	Healing agent leakage during cracking Mechanical property improvement Controllable quantity of healing agent	Complicated method and difficult to apply Needs verification for use in real projects Weakens the concrete and causes a decrease in mechanical properties with the usage of too many vessels
Passive hollow vessels	Healing agent leakage during cracking Mechanical property improvement	Complicated method and difficult to apply No sufficient data available for assessment The possible difficulty of releasing the healing agent
Encapsulation	Immediate response to cracking Less restoration of mechanical properties Medium amount of healing agent The capsule shell may not break due to insufficient magnitude of stresses created by cracking	Difficulty in preparing capsules Very limited amount of healing after filling the capsule A strong capsule shell is required to protect the healing agent during mixing and the hydration process The shell and matrix bond should be taken into consideration
Bacteria	Natural eco-friendly biological activity Mechanical property improvement Easy to implement Bacterial nutrients slow down the hydration process, resulting in lower strength	Leakage may occur during the mixing Measures should be taken for protection of bacteria

## Data Availability

No data, models, or code were generated or used during the study.

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
