# Peer review of "State-of-the-Art Report: The Self-Healing Capability of Alkali-Activated Slag (AAS) Concrete"

_materials, 2023, doi:10.3390/ma16124394_

Round 1
Reviewer 1 Report
This study is a review manuscript for the self-healing ability of AAS and its effect on the mechanical properties of AAS mortars. Several self-healing approaches, applications, and challenges of each mechanism are considered and compared between their impacts. The topic is worth investigating and the paper could be interesting to the readers of this journal. The authors are advised to revise the paper based on the following comments.
1、 Lines 287-289, 292, 507, 525, "Error! Reference source not found", please revise it.
2、 Is AAC an abbreviation for Alkali-activated composites? The title of this paper refers to "AASC", but the abbreviations AAC, AAS paste, AAS mortar, AAS concrete appear in many places in this paper, do they all refer to AASC?
3、 The title of Section 2.1 can be eliminated because there is no Section 2.2.
4、 The crack self-healing in this paper is mainly an apparent phenomenon, is there a review on the quantitative assessment of mechanical properties?
Author Response
The paper in its present form has been revised to address all the reviewer’s comments. The comments and the corresponding response for each of the reviewer’s comments are listed herein.

Reviewer 2 Report
The manuscript is an interesting overview material. There are the following comments:
1. It is necessary to correct the subscripts of CO2 in lines 16, 64, 65, 116.
2. In the equations of chemical reactions 2 and 3, the superscript in H+ must be corrected.
3. In lines 288, 292, 507, 525, instead of a link to the source, it says "Error! Reference source not found".
4. In Table 3 and equation 6, you need to correct the superscript for OH-.
5. In Table 4, there is no subscript at the end of the substance Ca(NO3) (it should be Ca(NO3)2).
6. Line 462 should have (NH4+) since it is an ion.
7. In equation 12, you need to correct the indexes on the right side of the expression.
8. In equation 13, the subscripts and superscripts need to be corrected.
9. In line 170, you need to correct the indexes in Na+ and OH-.
Author Response

(The authors gave the same response as above.)

Reviewer 3 Report
The paper was already submitted as a "student paper" to the University of West London (78% similarity). This aspect should be clarified before considering it for publication in MDPI.
On the other hand, the paper is a good review on self-healing capability of alkali-activated slag concrete, adding in a single place many useful information and details about materials and self-healing technologies available today.
The introductory part explains the need for this review, but the novelty is not very clear - it can be improved.
The main part, covering the "materials and methods" chapters, is well described, explained and referenced, but it needs major revision in terms of format for a proper understanding:
- all figures should be re-numbered... now they are mixed up... fig. 1 and 2 (lines 153 and 155) should be fig. 3 and 4; these figures also need more explanations (in short, just to avoid the need to look for the referenced paper); all the next figures are wrong numbered;
- many figures are unclear... the authors should try to replace them (fig. 6, fig. 3 - line 377, etc);
- all sub-chapters from chapter 3 should be re-ordered; after 3.3, comes 3.2.1.1 (line 358);
- use brackets () for equations;
- use proper writing style for chemical formulas: CO2 not CO2, CaCO3 not CaCO3, etc;
- there are a few missing references errors: line 288, 292, 507, etc.
Chapter 4 should be improved. This should be the "central point" of the paper according to the title.
Conclusions are fine and they can be a good starting point for future research and practical applications.
There are just minor grammar and spelling mistakes, easy to correct in the next revision.
Author Response

(The authors gave the same response as above.)
